# Wing Plasticity Is Associated with Growth and Energy Metabolism in Two Color Morphs of the Pea Aphid

**DOI:** 10.3390/insects15040279

**Published:** 2024-04-16

**Authors:** Hehe Cao, Xi Wang, Jiawei Wang, Zhaozhi Lu, Tongxian Liu

**Affiliations:** 1Shandong Engineering Research Center for Environment-Friendly Agricultural Pest Management, College of Plant Health and Medicine, Qingdao Agricultural University, Qingdao 266109, China; caohehe1988@163.com (H.C.); wangxii2024@163.com (X.W.); jiawei_wang333@163.com (J.W.); 2Institute of Entomology, Guizhou University, Guiyang 550025, China

**Keywords:** wing polymorphism, metabolism, glycolysis, lipolysis, reproduction

## Abstract

**Simple Summary:**

The pea aphid, *Acyrthosiphon pisum*, is a serious pest of legume crops that can produce winged and wingless offspring in response to the environmental conditions. We found that, under crowded conditions, the red morph *A. pisum* produced more winged offspring than the green morph. The signaling pathways involved in aphid wing determination also play important roles in regulating growth, development, and metabolism. Thus, we examined the association between the wing-producing ability and the growth rate, development time, reproductive capacity, and energy metabolism in these two color morphs. The growth rate, levels of glycogen and triglycerides in red morphs were significantly higher than those of green morphs, whereas green morphs produced more offspring. Red morphs also consumed more triglycerides during starvation and had stronger starvation tolerance. Furthermore, the expression levels of genes involved in the insulin pathway, glycolysis, and lipolysis in red aphids were also higher. These results suggest that the wing-producing ability of the pea aphid may be associated with its growth and metabolism.

**Abstract:**

The pea aphid, *Acyrthosiphon pisum*, is a major pest of legume crops, exhibiting distinct polymorphism in terms of wings and body color. We found that, under crowded conditions, the red morph *A. pisum* produced more winged offspring than the green morph. The signaling pathways involved in aphid wing determination, like insulin and ecdysone, also play important roles in regulating growth, development, and metabolism. Thus, here, we examined the association between the wing-producing ability and the growth rate, development time, reproductive capacity, and energy metabolism in these two color morphs. The growth rate of red morphs was significantly higher than that of green morphs, whereas green morphs produced more offspring during the first 6 days of the adult stage. Red morphs accumulated higher levels of glycogen and triglycerides and consumed more triglycerides during starvation; however, green aphids consumed more trehalose during food deprivation. Red aphids exhibited stronger starvation tolerance, possibly due to their higher triglyceride catabolic activity. Furthermore, the expression levels of genes involved in the insulin pathway, glycolysis, and lipolysis in red aphids were higher than those in green aphids. These results suggest that the wing-producing ability of the pea aphid may be associated with its growth and metabolism, which may be due to the shared regulatory signaling pathways.

## 1. Introduction

The pea aphid, *Acyrthosiphon pisum*, is a major pest of legume crops, feeding on the phloem sap of plants and causing significant economic losses. In recent years, the pea aphid has become a valuable model species in the study of genetics, ecology, and evolutionary biology because of its complex life cycle, diverse reproductive strategies, and remarkable phenotypic plasticity [1,2]. The pea aphid exhibits diverse phenotypic plasticity, including body color dimorphism (green vs. red) and wing dimorphism (winged vs. wingless) [2]. The pea aphid can parthenogenetically produce genetically identical winged and wingless offspring in response to environmental factors such as temperature, population density, and host plant nutrition [1]. The flight ability of winged aphids enables them to colonize new habitats and avoid resource competition, but at the cost of lower reproduction, thus creating a trade-off between reproduction and dispersal [3]. Therefore, understanding the mechanism underlying aphid wing dimorphism is crucial in predicting aphid migration and population dynamics.

The wing dimorphism of aphids is primarily regulated by their own genetics and environmental factors [4]. Crowding and host plant nutrition are two extensively studied environmental factors that influence aphid wing morph determination [4]. Crowding-related tactile stimuli and alarm pheromones are considered major factors inducing wing morph formation, although low host nutrition alone can increase the production of winged forms in some aphid species [1]. Several studies have investigated the molecular mechanisms that regulate aphid wing polymorphism. For example, the insulin and ecdysone signaling pathways have been reported to play a role in determining the wing dimorphism in the pea aphid [5,6]. Recently, microRNA-insulin signaling and the target of rapamycin (TOR) pathway have been identified as factors contributing to aphid wing plasticity [7,8]. However, different aphid species or asexual genotypes of the same aphid species exhibit different wing induction abilities in response to the same environmental cues, and the ultimate mechanism underlying aphid wing dimorphism remains elusive [1,9].

The pea aphid exhibits two color morphs, red and green, which have different genetic backgrounds and cannot be transformed by environmental changes [2]. These two color morphs show differences in dispersal, energy reserves, fecundity, and wing production [2,10]. Red morphs are generally more active and accumulate more lipids and carbohydrates than green morphs, but have lower reproductive capacity [10,11]. Additionally, the induction of winged offspring occurs more readily in red morphs than green morphs, while green morph clones differ in their ability to produce winged offspring [2,12]. The signaling pathways involved in the aphid wing induction process, namely insulin, ecdysone, and TOR, also play important roles in aphid growth, development, and metabolism, implying that these processes may indicate the activity of these signaling pathways and thus the wing-producing ability [5,6,7].

To investigate whether the aphid’s wing-producing ability was associated with its growth and metabolism, we first investigated the biological parameters of green and red morphs of pea aphids, including the wing induction ability, body weight gain, reproduction, and starvation tolerance. We then compared the differences in carbohydrate, lipid, amino acid, and protein content in these two colored aphids before and after crowding and starvation treatment. Furthermore, we analyzed the expression of genes involved in insulin signaling and energy metabolism. Understanding the interaction between aphid growth, metabolism, and their wing induction ability will provide insights into the mechanism behind the ecological adaptation of aphids to changing environments.

## 2. Materials and Methods

### 2.1. Plants and Insects

The red and green morph pea aphids were initially established from a respective single viviparous adult that was collected from *Medicago sativa* L. at Gansu Agricultural University (Anning District, Lanzhou, China) in 2015. The pea aphids were reared on *Vicia faba* plants that were grown in plastic pots (12.5 cm in diameter) with a garden soil mixture. The plants and aphids were reared in climate incubators set at 21 °C, 14 L:10 D, and 70 ± 10% RH.

### 2.2. Aphid Wing Induction

Fifteen wingless adult aphids (3–5-d-old adults) of the respective color morph were collected and placed in a 35 × 15 mm Petri dish, where they were starved and crowded for 0, 6, 12, or 24 h. Subsequently, every three adults were transferred to one mature *V. faba* leaf, confined by a nylon mesh bag as one replicate [13]. After 24 h, adult aphids were removed, and the wing morphs of the remaining nymph were examined when they developed into adults.

### 2.3. Aphid Growth, Fecundity, and Starvation Tolerance

Two red or green apterous adult aphids were placed on one mature *V. faba* leaf and confined by a nylon mesh bag. After 12 h, adult aphids were removed, leaving five nymphs on each leaf. Then, the weight of the nymphs on each leaf was measured by a microbalance (Sartorius MSA 3.6 P-000-DM, resolution 0.001 mg, Gottingen, Germany) after different development times. All aphids measured were apterous, as alate nymphs were excluded in this assay. Nymphs on each leaf were regarded as one replicate and eight replicates were performed for each color morph.

To measure the fecundity of the pea aphid, one newly molted red or green apterous adult aphid was confined on a *V. faba* leaf by a nylon mesh bag. The number of offspring produced was counted and removed every two days for the first six days of the adult period, with eighteen replicates for each color morph.

To assess the starvation tolerance of the pea aphid, about 30 green or red adult aphids were individually placed in the cells of 24-well plates. We conducted the experiment under room humidity or with 2% agar in each cell to maintain high humidity. The number of surviving aphids was recorded every 12 h until all aphids had died.

### 2.4. Metabolite Extraction and Analysis

We collected and measured the weights of 2-d-old wingless adult aphids that had fed on *V. faba* leaves or been subjected to wing induction conditions (crowding and starvation treatment for 12 h). Amino acids and sugars were extracted and then analyzed by an LTQ-XL linear ion trap mass spectrometer (Thermo Scientific, Waltham, MA, USA), as described in Cao et al. [13] and Ahmed et al. [14]. The glycogen content was determined using the anthrone sulfate method, using a glycogen content test kit [15] (BC0345; Solarbio Science & Technology Co., Ltd., Beijing, China), while the triglyceride content was determined using a triglyceride content assay kit (BC0620; Solarbio Science & Technology Co., Ltd., Beijing, China). We used the Bradford method to determine the protein content, as described in our previous publication [13].

### 2.5. Gene Expression Analysis

RNA-seq was performed according to our previous publication [16]. Two-day-old nymphs (second instar) were collected, immediately frozen in liquid nitrogen, and stored at −80 °C in a refrigerator until use. Three biological replicates were performed for each color aphid, with each replicate consisting of 30 nymphs. Total RNA was extracted using RNAiso Plus (Takara Biotechnology, Dalian, China), following the manufacturer’s protocol. The RNA integrity was assessed using the Bioanalyzer 2100 system (Agilent Technologies, City of Santa Clara, CA, USA). RNA meeting the quality criteria was sent to Novogene Co., Ltd. (Beijing, China) for library preparation and transcriptome sequencing. Transcriptome analysis was conducted based on our previous study [16]. Differential expression genes were identified using DESeq2, with genes having adjusted *p*-values < 0.05 considered differentially expressed genes. The raw reads of the RNA-seq were submitted to NCBI’s Sequence Read Archive. The accession numbers for these sequences are GenBank: SRX24097866-SRX24097871. Heatmaps of the differentially expressed genes were constructed with the binary log of the fold change of red/green for each gene in the TBtools software v2.061 [17].

We then used quantitative real-time PCR analysis (qRT-PCR) to examine the expression of selected insulin and metabolic genes in adult aphids subjected to crowding and starvation treatment. The green or red morph adults feeding on *V. faba* leaves were crowded and starved in a 35 × 15 mm Petri dish for 12 h in groups of 15 aphids. Then, ten adult aphids were collected for each replicate, rapidly frozen in liquid nitrogen, and stored at −80 °C. We used the RNAiso Plus (Takara Biotechnology, Dalian, China) to extract the total RNA, following the manufacturer’s instructions. One μg of RNA was used to synthesize cDNA with the PrimeScript^TM^ RT Reagent Kit (TaKaRa Biotechnology, Dalian, China). The cDNA was then diluted 50 times with purified water and used as templates for qRT-PCR with primers specific to the target gene (Table 1). The qRT-PCR assay was carried out using a LightCycler^®^ 96 Instrument (Roche Diagnostics, Basel, Switzerland) and the Takara SYBR^®^ PremixExTaq under the following conditions: 95 °C for 30 s, followed by 40 cycles of 95 °C for 10 s and 58 °C for 20 s. The *NADH* gene was used as the reference gene [18]. The gene amplification efficiency was determined by a series of 5-fold diluted cDNA, and the amplification specificity was determined by melt curve analysis (Appendix A). The relative gene expression levels were calculated using the 2^−∆∆CT^ method [19]. Four biological replicates were performed for each treatment, with three technical replicates for each biological replicate.

### 2.6. Statistical Analysis

Two group comparisons were analyzed by the unpaired Student’s *t*-test, while Kolmogorov–Smirnov tests were used to test the normality of the data before analysis. The log-rank test was used to analyze aphid survival in the starvation tolerance assay. We considered *p* < 0.05 statistically significant. Statistical analyses were performed using GraphPad Prism 8.0.2 and the IBM SPSS Statistics Package 19 (SPSS Inc., Chicago, IL, USA).

## 3. Results

### 3.1. Wing Production, Growth, Reproduction, and Starvation Tolerance

The proportion of winged offspring produced by both color aphids reared in isolation was near zero. However, when subjected to crowding and starvation treatment for 6, 12, or 24 h, red aphids produced a significantly higher percentage of winged offspring than green aphids (Student’s *t*-test, *p* < 0.05; Figure 1A). In contrast, the proportion of winged offspring produced by green morphs consistently remained below 5% (Figure 1A).

To assess the growth rate of the aphids, we recorded the weight of the pea aphids at different developmental stages. At 0 d post-birth, the body weight between the two color morph aphids was similar (*t* = 0.001, df = 12, *p* > 0.99; Figure 1B). However, on the 2nd (second instar), 4th (third instar), and 8th days (adult), the weight of the red pea aphids was significantly higher than that of the green pea aphids (Student’s *t*-test, *p* < 0.05; Figure 1B), indicating that the red aphids had a higher growth rate. Moreover, the red aphids took a longer time to develop into adult aphids from birth (*t* = 2.275, df = 34, *p* = 0.029; Figure 1C).

During the first two days, the number of offspring produced by both color aphids was similar (*t* = 0.72, df = 34, *p* = 0.48; Figure 1D). However, on the 3rd–4th and 5th–6th days, the number of offspring produced by red pea aphids was significantly lower (Student’s *t*-test, *p* < 0.05; Figure 1D). The total number of offspring produced by red pea aphids during the first 6 days was also significantly lower than that of green pea aphids (*t* = 2.99, df = 33, *p* < 0.05; Figure 1D). However, the red aphids were able to survive longer under starvation than the green pea aphids under both high humidity and dry conditions (log-rank test, *p* < 0.01; Figure 1E,F).

### 3.2. Energy Metabolites in Aphids before and after Starvation

We found that the glycogen (*t* = 3.575, df = 14, *p* = 0.003; Figure 2A) and triglyceride (*t* = 2.874, df = 14, *p* < 0.05; Figure 2B) levels in the red pea aphids were significantly higher than those in the green pea aphids. After crowding and starvation for 12 h, both color aphids exhibited a decrease in glycogen and triglyceride levels. Notably, the decrease in triglycerides was greater in red pea aphids (9.6 μg/mg) than in green pea aphids (5.3 μg/mg) (*t* = 2.424, df = 14, *p* < 0.05; Figure 2C). There was no significant difference in the total amount of amino acids, glucose, trehalose, or protein in both color aphids (Figure 2D–F,H), while the decrease in trehalose was greater in green pea aphids than in red pea aphids (*t* = 2.269, df = 14, *p* < 0.05; Figure 2G) after 12 h of starvation.

### 3.3. Expression Analysis of Genes Involved in Insulin Pathway and Energy Metabolism

To obtain an overview of the differential gene expression in red and green morph aphids, principal component analysis (PCA) was performed. Principal component axis 1 explained ~51% of the variance, indicating a large effect of the aphid color on gene expression. Principle component axis 2, with 15% of the variance, suggested that different samples within each color aphid had small variance (Appendix A).

Compared with the green morph aphids, the first-instar red morph aphids exhibited higher expression levels of insulin signaling genes *insulin receptor 1* (*InR1*), *insulin receptor 2* (*InR2*), *protein kinase B* (*AKT*), and *forkhead transcription factor subgroup* O (*FOXO*), while the expression level of *insulin 4* was lower (*p* < 0.05; Figure 3A). Additionally, the red morph aphids displayed significantly higher expression levels of glycolysis genes *hexokinase* (*HK*), *glucose-6-phosphate isomerase* (*PGI*), and *phosphofructokinase-1* (*PFK-1*), as well as lipolysis genes *adipokinetic hormone receptor* (*AKHR*), *carnitine O-palmitoyltransferase 1* (*CPT1*), *carnitine O-palmitoyltransferase 2* (*CPT2*), *lipid storage droplet protein 1* (*LSD1*), *lipase 3, apolipophorin-1/2*, and *lipophorin receptor* (*LpR*), compared with the green pea aphids; however, red aphids exhibited a lower expression level of the glycolysis gene enolase (*p* < 0.05; Figure 3B,C). All differentially expressed genes between red and green morphs can be found in Appendix A.

There was no significant difference in the expression of insulin and metabolism genes between red and green morph adult aphids, while both groups exhibited an increase in *HK* expression following crowding and starvation treatment (*p* < 0.05; Figure 4). However, after such treatment, the *PFK-1* gene expression in green morph aphids was higher than that in red morphs, while the gene expression of *CPT1* and *InR2* in green morphs was lower (*p* < 0.05; Figure 4). Additionally, green morphs subjected to crowding treatment exhibited an increase in *InR1* expression, but a decrease in *AKT* gene expression (*p* < 0.05; Figure 4).

## 4. Discussion

In this study, we observed a significant increase in the production of winged offspring in red morph pea aphids under conditions of crowding and starvation, while the proportion of winged offspring in green morph aphids consistently remained below 5%. This result is consistent with previous studies, which found that red morph pea aphids were generally more able to induce winged offspring, while green morph clones varied in their wing induction abilities [2,12]. The growth rate and metabolism activity of red morph pea aphids were higher than those of green morphs, whereas their reproduction ability was lower. Therefore, our results suggest that higher growth and metabolic rates may be associated with a stronger ability to produce winged offspring in aphids, but at the cost of reduced reproductive potential. These findings align with the dispersal–reproduction trade-off hypothesis, which suggests that an insect’s flight capability interacts antagonistically with their fecundity [3].

Flight in insects is an energetically demanding state that requires the rapid consumption of large amounts of energy [20]. Therefore, aphids need the ability to store and rapidly utilize energy for flight [20]. In this study, red morph aphids exhibited a faster body weight gain, higher levels of energy reserves such as glycogen and TAG, and faster TAG consumption during starvation, all of which are advantageous for flight. Additionally, red morph aphids showed greater resistance to starvation compared with green aphids, indicating their higher energy utilization efficiency during food deprivation [21]. The fact that the winged offspring of red morphs were only produced when the aphids were under starvation and high-density conditions implies a potential relationship between catabolism and aphid wing induction. This finding is consistent with a previous study that found a positive correlation between the resting metabolic rates of different genotypes of pea aphids and their production of winged offspring [22]. In turn, this higher energy expenditure in red aphids may also contribute to their lower reproduction capacity.

The insulin signaling pathway plays an important role in glucose metabolism, lipid and glycogen synthesis, and overall biological growth [23]. In this study, we observed that the expression levels of insulin-related genes, namely *InR1*, *InR2*, and *AKT*, were higher in the first instar of red morph aphids compared with green morphs. Furthermore, red morphs exhibited faster growth and higher levels of glycogen and TAG, all of which are regulated by insulin signaling. These findings suggest that insulin signaling is more active in red morph aphids than in green morph aphids. Because insulin signaling is known to determine wing morph switching in planthoppers and is also involved in wing dimorphism in brown citrus aphids and pea aphids, the higher insulin activity in red aphid nymphs may contribute to their higher proportion of winged offspring [6,7,24].

However, when adult aphids were exposed to wing induction conditions (crowding and starvation), both the glucose and trehalose levels decreased, suggesting that insulin, which promotes anabolic metabolism after food ingestion, may not be a direct factor involved in wing induction in adult aphids. Instead, under such conditions, aphids exhibit the upregulation of genes involved in glucose and lipid catabolism, leading to accelerated catabolic activity. Because flight requires higher energy metabolism potential, these results suggest that catabolism in adult aphids during wing induction may be associated with the wing induction ability [22]. Similarly, artificially stimulating individual starved aphids increased their movement and thus catabolic metabolism, coinciding with a higher proportion of winged offspring than in unstimulated aphids [12,25]. These results suggest that catabolic metabolism activity may play a role in wing induction. Nevertheless, further research is necessary to elucidate the relationship between insulin signaling, metabolism, and wing induction in aphids.

In conclusion, our findings suggest that both the anabolism (growth) and catabolism of pea aphids may be associated with their ability to produce winged offspring, which nevertheless needs further validation in more aphid genotypes or aphid species. The higher metabolic rate of red aphids may enable them to better tolerate adverse environmental stresses or escape from unfavorable conditions by producing winged offspring. In contrast, green aphids appear to maximize their fitness by allocating more energy and resources to reproduction. Our results suggest that pest management strategies targeting aphid metabolism may affect aphids’ adaptation to adverse environments, their migration, and their reproduction.

## Figures and Tables

**Figure 1 insects-15-00279-f001:**
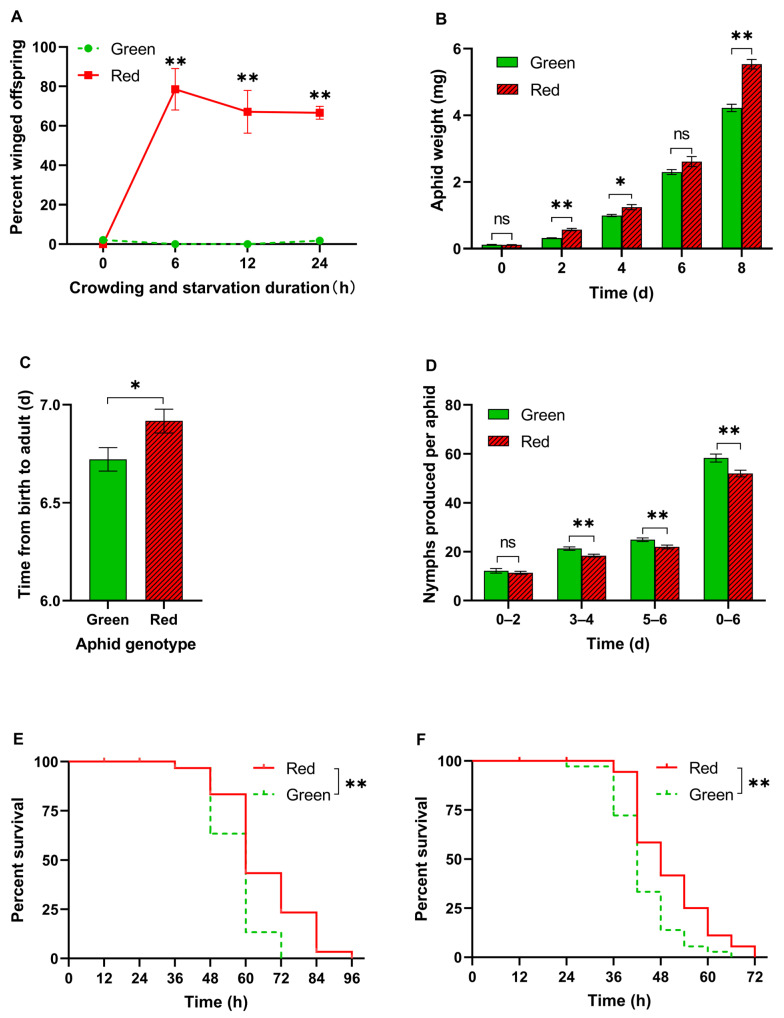
Biological parameters of red and green aphids. (**A**) Percentage of winged offspring after different crowding and starvation periods (Student’s *t*-test, *n* = 6). (**B**) Weight of pea aphids at different development times (Student’s *t*-test, *n* = 8). (**C**) Development time from birth to adults (Student’s *t*-test, *n* = 18). (**D**) Number of offspring produced by each aphid in the first 6 days of adulthood (Student’s *t*-test, *n* = 18). (**E**,**F**) Starvation tolerance under high humidity (**E**) or room conditions (**F**) (log-rank test, *n* = 30). Data are shown as mean ± SEM. * *p* < 0.05, ** *p* < 0.01. ns: no significance.

**Figure 2 insects-15-00279-f002:**
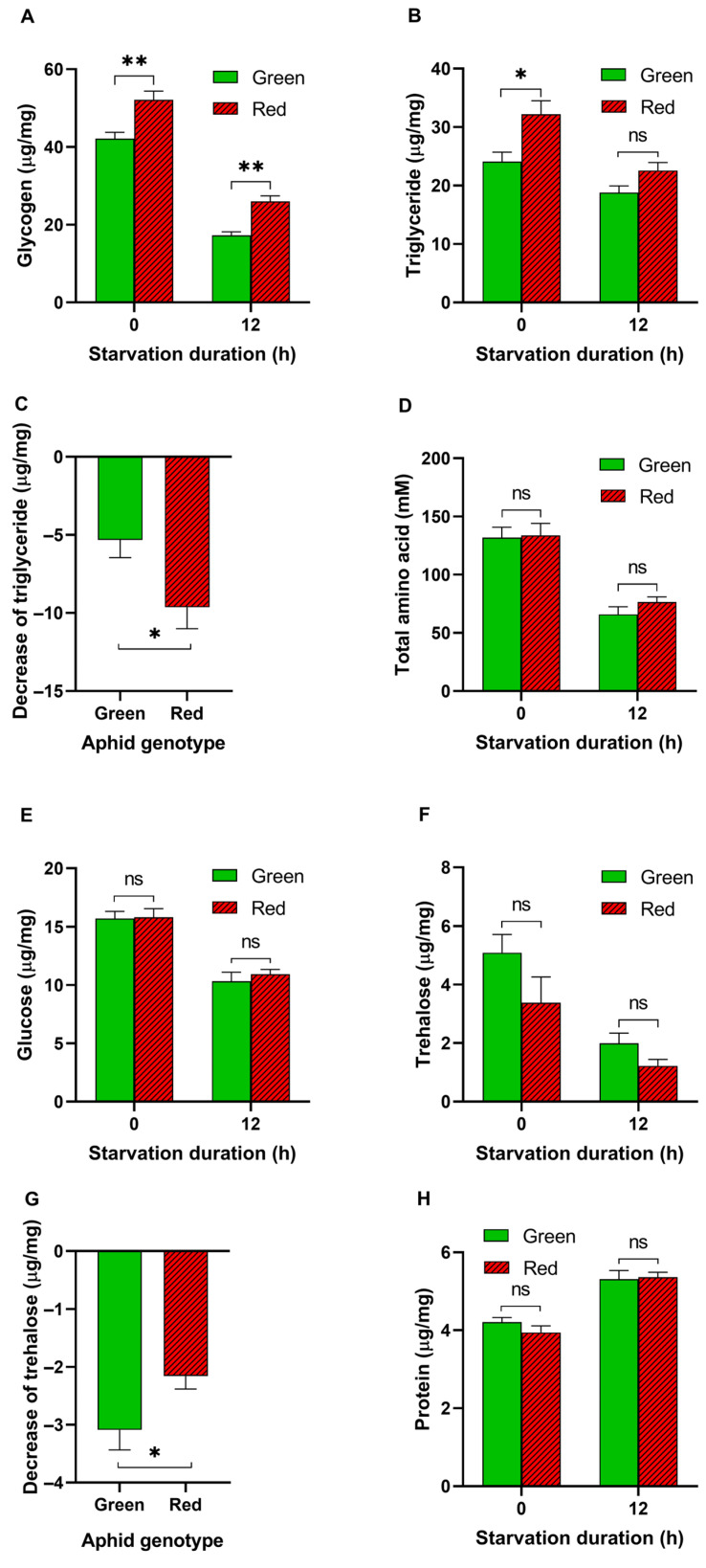
Energy metabolites in aphid body before and after starvation. (**A**–**C**) Glycogen level (**A**), triglyceride level (**B**), and decrease in triglycerides after starvation for 12 h (**C**) in aphids. (**D**–**F**) Total amino acid level (**D**), glucose level (**E**), and trehalose level (**F**). (**G**) Decrease in trehalose after 12 h starvation. (**H**) Total protein level. (Student’s *t*-test, *n* = 8, * *p* < 0.05, ** *p* < 0.01. ns: no significance). Data are shown as mean ± SEM.

**Figure 3 insects-15-00279-f003:**
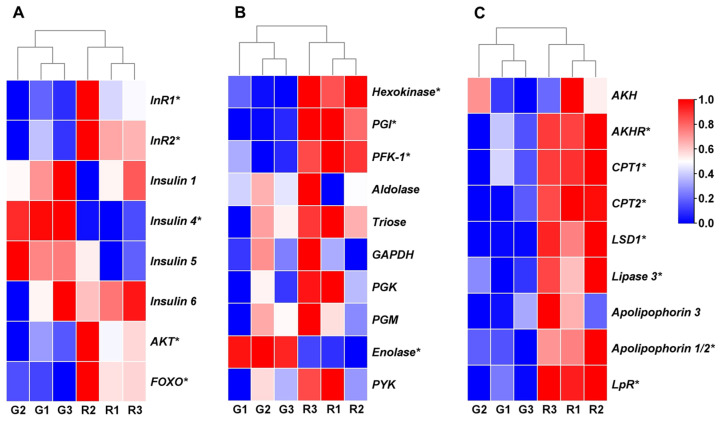
Expression of genes involved in insulin pathway and energy metabolism. (**A**) Heatmap showing genes involved in insulin pathway in pea aphids. (**B**,**C**) Heatmap representing expression levels of genes related to glycolysis (**B**) and lipolysis (**C**) in pea aphids. Heatmap representing log2 fold change values between red and green aphids. Scale method: zero to one. Red color represents higher expression, while blue represents lower expression. Student’s *t*-test, *n* = 3, * *p* < 0.05. Abbreviations: G1–3: Green aphid replicate 1–3; R1–3: Red aphid replicate 1–3; *InR1*: *insulin receptor 1*; *InR2*: *insulin receptor 2*; *AKT*: *protein kinase B*; *FOXO*: *forkhead transcription factor subgroup O*; *HK*: *hexokinase*; *PGI*: *glucose-6-phosphate isomerase*; *PFK-1*: *phosphofructokinase-1*; *GAPDH*: *glyceraldehyde-3-phosphate dehydrogenase*; *PGK*: *phosphoglycerate kinase*; *PGM*: *phosphoglycerate mutase*; *PYK*: *pyruvate kinase*; *AKH*: *adipokinetic hormone*; *AKHR*: *adipokinetic hormone receptor*; *CPT1*: *carnitine O-palmitoyltransferase 1*; *CPT2*: *carnitine O-palmitoyltransferase 2*; *LSD1*: *lipid storage droplet protein 1*; *LpR*: *lipophorin receptor*.

**Figure 4 insects-15-00279-f004:**
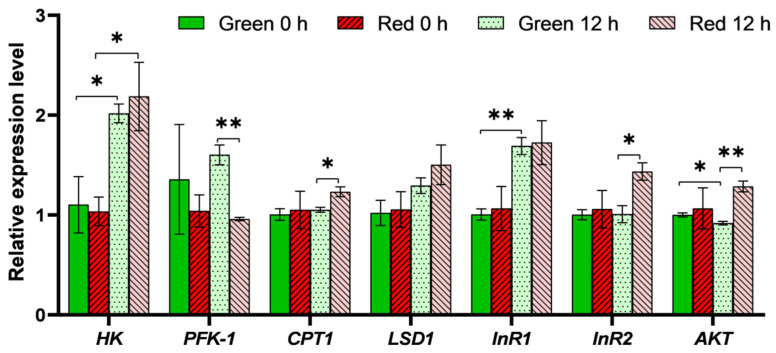
Expression of selected insulin and metabolic genes in red and green adult aphids before and after crowding treatment. Student’s *t*-test, *n* = 3, * *p* < 0.05, ** *p* < 0.01. Pairwise comparisons between aphid color or treatment time without asterisks were not significantly different. Data are shown as mean ± SEM. Abbreviations: *HK*: *hexokinase*; *PFK-1*: *phosphofructokinase-1*; *CPT1*: *carnitine O-palmitoyltransferase 1*; *LSD1*: *lipid storage droplet protein 1*; *InR1*: *insulin receptor 1*; *InR2*: *insulin receptor 2*; *AKT*: *protein kinase B*.

**Table 1 insects-15-00279-t001:** Primers for real-time quantitative polymerase chain reaction.

Primer Name	Primer Sequences (5′ to 3′)	GenBank Accession Number
*NADH*-Forward	CGAGGAGAACATGCTCTTAGAC GATAGCTTGGGCTGGACATATAG	NM_001162323.2
*NADH*-Reverse		
*HK*-Forward	TGTACATGGGCGAGATCGTG	XM_003242192.4
*HK*-Reverse	GGATACGAATGACGTGCCGA	
*PFK-1*-Forward	ACCGTGATCCGTGACCTCTA	XM_029487259.1
*PFK-1*-Reverse	TCATGTTCAGCCCTGGCAAT	
*CPT1*-Forward	CTGAGCATTCGTGGGCTGAT	XM_029488698.1
*CPT1*-Reverse	GGAGGTTCCAATTCAGGAGCA	
*LSD1*-Forward	GCCGACAAACCGAGTACAGA	XM_003240065.4
*LSD1*-Reverse	CACGAAGCACTGGACGGTAT	
*InR1*-Forward	ATCTGTCCACCGGAATGTGG	MK510961.1
*InR1*-Reverse	TCGGCTGGACATTCCTTCAC	
*Insulin R2*-Forward	CGAGCCTCCTGAAATCACGTA	XM_008184754.3
*Insulin R2*-Reverse	TGGCATTTTTGTCGTCCTTGTG	
*AKT*-Forward	TGGCTGTTCAAACGAGGAGA	XM_008185246.3
*AKT*-Reverse	CGGTCATTGGACCTTCTGGT	

## Data Availability

The data presented in this study are available on request from the corresponding author.

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
