# Peer review of "Wing Plasticity Is Associated with Growth and Energy Metabolism in Two Color Morphs of the Pea Aphid"

_insects, 2024, doi:10.3390/insects15040279_

Round 1

Reviewer 1 Report

Comments and Suggestions for Authors

The authors present the analysis of different characteristics of two color morphs of the pea aphid Acyrthosiphon pisum, which produce different proportions of winged and wingless offspring in response to different environmental conditions.

Major comments:

The analysis of the life history traits is well described and convincing. I have only one comment on the trehalose level. Figure 2F shows a difference between green aphids and red aphids, so I am surprised that the trehalose level is not significantly different between them, especially since the decrease in trehalose after starvation is higher in green aphids than in red aphids (figure 2G). 

The differential analysis of gene expression raises a number of questions. It is not clear whether the data presented in Figure 3 represent all genes differentially expressed between green and red aphids? A table showing the complete data set should be included in the supplementary data. In addition, the RNA-seq data should be deposited in a public database and made available to reviewers. What do G1, G2 and G3, R1, R2 and R3 correspond to? Are there biological replicates? Some triplicates look very different. The authors should do a PCA analysis to show how the three samples cluster. More samples would be needed to confirm the results.

qRT-PCR (Figure 4) must follow the MIQE guidelines (Bustin et al., Clinical Chemistry, 2009). In particular, to be able to compare the expression of the target gene with that of the NADH reference gene using the 2DDCTmethod, the efficiency of each pair of primers should be approximately 2, and this must be reported in the material and methods.

In Figure 4, is the difference of HK expression between green aphids and red aphids after crowding significant? Is the difference in InR1 expression between red aphids before and after crowding significant? Same question for InR2 and AKTAKT expression decreases in green aphids after crowding. Could the authors comment on this result?

Minor comments:

- Line 129: repeat of “the”

- Figure 1C: the difference between green and red aphids would be more visible if the scale of the y-axis was logarithmic.

- Gene names should be in italics.

Comments on the Quality of English Language

English is fine.

Author Response

Responses to reviewer 1

Comments 1. I have only one comment on the trehalose level. Figure 2F shows a difference between green aphids and red aphids, so I am surprised that the trehalose level is not significantly different between them, especially since the decrease in trehalose after starvation is higher in green aphids than in red aphids (figure 2G).

Response 1: we performed statistically analysis for figure 2f and did not find significant difference between green aphids and red aphids both at 0 h and 12 h, which may be explained by the large variance of trehalose level within each color aphid. Because glucose is the direct sugar used by aphid metabolism, the content of trehalose in aphid bodies is not as stable as glucose. The decrease of trehalose for green aphids is higher than red aphids, which may be due to a larger trehalose consumption by green aphids.

Comments 2. The differential analysis of gene expression raises a number of questions. It is not clear whether the data presented in Figure 3 represent all genes differentially expressed between green and red aphids? A table showing the complete data set should be included in the supplementary data.

Response 2. Figure 3 represents Expression of genes involved in insulin pathway and energy metabolism. In this study we focus on insulin, sugar and lipid metabolism that involved in growth and wing dimorphism. We have upload all differentially expressed genes as supplementary data as suggested.

Comments 3. The RNA-seq data should be deposited in a public database and made available to reviewers.

Response 3. We have deposit the RNA-seq data to NCBI’s Sequence Read Archive. The accession numbers for these sequences are GenBank: SRX24097866-SRX24097871 (Line 154-156).

Comments 4. What do G1, G2 and G3, R1, R2 and R3 correspond to?

Response 4. We have indicated G and R in figure 3 legend as follows. G1-3: Green aphid replicate 1-3; R1-3: Red aphid replicate 1-3.

Comments 5. Are there biological replicates? Some triplicates look very different. The authors should do a PCA analysis to show how the three samples cluster. More samples would be needed to confirm the results.

Response 5. Yes, we performed three biological replicates for RNA-seq analysis. We have made a PCA analysis and found that principal component axis 1 explained ~51% of the variance, indicated a large effect of aphid color on gene expression. Principal component axis 2 exhibits 15% of the variance, suggested that different samples within each color aphid had a small variance (line 238-242). The PCA figure can be found in supplementary data.

Comments 6. qRT-PCR (Figure 4) must follow the MIQE guidelines (Bustin et al., Clinical Chemistry, 2009). In particular, to be able to compare the expression of the target gene with that of the NADH reference gene using the 2- DDCTmethod, the efficiency of each pair of primers should be approximately 2, and this must be reported in the material and methods.

Response 6. We performed experiments to test the amplification efficiency and specificity and found that the gene amplification efficiency is 0.911 to 0.966. “Gene amplification efficiency was determined by a series of 5-fold diluted cDNA, and amplification specificity was determined by melt curve analysis” (Line 179-181). The amplification efficiency and specificity can be found in supplementary materials.

Comments 7. In Figure 4, is the difference of HK expression between green aphids and red aphids after crowding significant? Is the difference in InR1 expression between red aphids before and after crowding significant? Same question for InR2 and AKT? AKT expression decreases in green aphids after crowding. Could the authors comment on this result?

Response 7. Yes, we have marked all significant difference in figure 4. We indicated this in figure 4 legend as “Pairwise comparisons between aphid color or treatment time without asterisks were not significantly different”.

Comments 8. Minor comments: - Line 129: repeat of “the”

- Figure 1C: the difference between green and red aphids would be more visible if the scale of the y-axis was logarithmic.

- Gene names should be in italics.

Response 8. We have deleted the repeat “the”.

Figure 1C we changed the start of Y-axis and now it is visible. We found that logarithmic change was not clear and direct.

We have changed gene name in manuscript and figures as suggested.

Reviewer 2 Report

Comments and Suggestions for Authors

Q.1 In the wing induction experiment, the authors observed a higher proportion of winged offspring in red morph aphids compared to green morphs under conditions of crowding and starvation. Could you elaborate on the underlying mechanisms driving this observed difference in wing induction between the two morphs? Are there specific genetic pathways or physiological factors that might explain this disparity?

Q. 2 The results show significant differences in growth rates, fecundity, and starvation tolerance between red and green aphids. Considering the observed variations in energy metabolites and gene expression related to insulin signaling and metabolism, how do these findings contribute to our understanding of the adaptive strategies employed by aphids under different environmental conditions?

Q. 3 The methodology for metabolite extraction and analysis involved various techniques such as mass spectrometry and biochemical assays. Could the authors elaborate on the reliability and reproducibility of these methods, particularly in the context of measuring glycogen, triglyceride, and protein content in aphids?

Q. 4 Considering the comprehensive approach used to characterize aphid physiology and gene expression profiles, how do the findings of this study contribute to our broader understanding of insect adaptation to environmental stressors and their implications for agricultural pest management strategies?

Q. 5 Given the observed decrease in glucose and trehalose levels in adult aphids under wing-induction conditions, could future studies explore the specific metabolic pathways or enzymatic activities responsible for this shift towards catabolic metabolism, potentially through metabolomic profiling or targeted enzyme inhibition assays?

Comments on the Quality of English Language

Need to be improved.

Author Response

Responses to reviewer 2

Comment 1. Q.1 In the wing induction experiment, the authors observed a higher proportion of winged offspring in red morph aphids compared to green morphs under conditions of crowding and starvation. Could you elaborate on the underlying mechanisms driving this observed difference in wing induction between the two morphs? Are there specific genetic pathways or physiological factors that might explain this disparity?

Response 1. Thank you for your insightful question. The underlying mechanisms driving this difference in wing induction could be multifaceted and involve a combination of genetic pathways and physiological factors. One possible explanation for the disparity in wing induction between the two morphs could be related to the insulin signaling pathway that regulate wing dimorphism in planthoppers and aphids. Results from gene expression suggested that red morph aphids possess higher insulin signaling.

We discuss this in Line 313-321: “In this study, we observed that the expression levels of insulin-related genes, namely InR1, InR2, and AKT, were higher in the first instar of red morph aphids compared with green morphs. Furthermore, red morphs exhibited a faster growth, higher levels of glycogen and TAG, all of which are regulated by insulin signaling. These findings suggest that the insulin signaling is more active in red morph aphids than in green morph aphids. Because insulin signaling is known to determine wing morph switching in planthoppers and is also involved in wing dimorphism in brown citrus aphid and pea aphids, the higher insulin activity in red aphid nymphs may contribute to their higher proportion of winged offspring.”

Comment 2. Q. 2 The results show significant differences in growth rates, fecundity, and starvation tolerance between red and green aphids. Considering the observed variations in energy metabolites and gene expression related to insulin signaling and metabolism, how do these findings contribute to our understanding of the adaptive strategies employed by aphids under different environmental conditions?

Response 2. The differences in growth rates, fecundity, and starvation tolerance between red and green aphids suggest that red aphids have higher insulin signaling that contribute to higher growth and higher metabolic rate, which may enhance red aphids to adapt to environmental stresses. However, high metabolic rate is costly and thus may reduce energy and metabolites for reproduction.

We discuss this in Line 289-295: “The growth rate and metabolism activity of red morph pea aphids were higher than those of green morphs, whereas their reproduction ability was lower. Therefore, our results suggest that higher growth and metabolic rates may associate with a stronger ability to produce winged offspring in aphids, but at the cost of reduced reproductive potential. These findings align with the dispersal-reproduction trade-off hypothesis, which suggests that the insect’s flight capability is antagonistically interacting with their fecundity.” And line 338-342, “The higher metabolic rate of red aphids may enable them to better tolerate adverse environmental stresses or escape from unfavorable conditions by producing winged off-spring. In contrast, green aphids appear to maximize their fitness by allocating more energy and resource to reproduction.”

Overall, these findings contribute to our understanding of how aphids adapt to different environmental conditions.

Comment 3. Q. 3 The methodology for metabolite extraction and analysis involved various techniques such as mass spectrometry and biochemical assays. Could the authors elaborate on the reliability and reproducibility of these methods, particularly in the context of measuring glycogen, triglyceride, and protein content in aphids?

Response 3. To ensure the accuracy and consistency of our methods, we validated and optimized each step of the metabolite extraction and analysis process. For metabolite extraction, we followed established protocols that have been widely used and validated. Mass spectrometry is a highly sensitive and reliable technique for quantifying metabolites, and we have many years of experience and published related papers.

Moreover, we conducted rigorous quality control measures to minimize variability. This included using a series of different concentration of external standards and performing technical replicates to ensure the accuracy and reliability of our measurements.

Comment 4. Q. 4 Considering the comprehensive approach used to characterize aphid physiology and gene expression profiles, how do the findings of this study contribute to our broader understanding of insect adaptation to environmental stressors and their implications for agricultural pest management strategies?

Response 4. Understanding how aphids respond to stressors such as temperature fluctuations, food availability, and pesticide exposure is crucial for developing effective pest management strategies. Our study suggests that metabolism may play a role in aphids overcoming these challenges. We discuss this in Line 338-342, “The higher metabolic rate of red aphids may enable them to better tolerate adverse environmental stresses or escape from unfavorable conditions by producing winged off-spring. In contrast, green aphids appear to maximize their fitness by allocating more energy and resource to reproduction.”

In addition, “our results suggest that pest management strategies targeting aphid metabolism may affect aphid adaptation to adverse environments, migration, and reproduction.”(Line 342-344)

Comment 5. Q. 5 Given the observed decrease in glucose and trehalose levels in adult aphids under wing-induction conditions, could future studies explore the specific metabolic pathways or enzymatic activities responsible for this shift towards catabolic metabolism, potentially through metabolomic profiling or targeted enzyme inhibition assays?

Response 5. The observed difference in decrease of triacylglycerol, glucose, and trehalose levels in adult aphids under wing-induction conditions warrants further investigation into the specific metabolic pathways or enzymatic activities responsible for this shift towards catabolic metabolism. Future studies could explore this by conducting metabolomic profiling and targeted enzyme inhibition assays. Nevertheless, insect metabolism is intricate and involves various signaling pathways. This study specifically investigates the relationship between wing plasticity, growth, and energy metabolism in the pea aphid. Our findings reveal that the red morph aphid demonstrates elevated activity in insulin, glycolysis, and lipolysis. Therefore, we did not discuss too much about these contents.

Round 2

Reviewer 1 Report

Comments and Suggestions for Authors

The authors have responded correctly to my comments and amended the text accordingly.

Reviewer 2 Report

Comments and Suggestions for Authors

Thank you for addressing the comments and suggestions I provided on the manuscript titled "Wing plasticity associates with growth and energy metabolism in two color morphs of the pea aphid" The authors have provided valuable clarifications and additional information. 

Comment 1: I am satisfied with the authors response. Thank you for the detailed explanation. It's clear how the higher insulin signaling in red morph aphids could contribute to the observed difference in wing induction. 

Comment 2:  I am satisfied with the authors response. Your response provides valuable insights into the adaptive strategies of aphids under different environmental conditions. 

Comment 3: Thank you for addressing the concerns regarding method reliability and reproducibility. This enhances confidence in the accuracy of the metabolite extraction and analysis techniques used in the study.

Comment 4: I am satisfied with the authors response.

Comment 5: Your acknowledgment of the need for further investigation into the specific metabolic pathways responsible for the observed changes in glucose and trehalose levels is noted. 

Overall, your responses provide a comprehensive and insightful elaboration on the key points raised, enhancing the clarity and significance of your study's findings.